# The Diagnostic and Prognostic Value of a Liquid Biopsy for Esophageal Cancer: A Systematic Review and Meta-Analysis

**DOI:** 10.3390/cancers12103070

**Published:** 2020-10-21

**Authors:** Daisuke Matsushita, Takaaki Arigami, Keishi Okubo, Ken Sasaki, Masahiro Noda, Yoshiaki Kita, Shinichiro Mori, Yoshikazu Uenosono, Takao Ohtsuka, Shoji Natsugoe

**Affiliations:** 1Department of Digestive Surgery, Breast and Thyroid Surgery, Kagoshima University Graduate School of Medical and Dental Sciences, Kagoshima 890-8520, Japan; ok0627@m2.kufm.kagoshima-u.ac.jp (K.O.); k-sasaki@m.kufm.kagoshima-u.ac.jp (K.S.); nodamasahiro69@kufm.kagoshima-u.ac.jp (M.N.); north-y@m.kufm.kagoshima-u.ac.jp (Y.K.); morishin@m3.kufm.kagoshima-u.ac.jp (S.M.); takao-o@kufm.kagoshima-u.ac.jp (T.O.); 2Department of Onco-biological Surgery, Kagoshima University Graduate School of Medical and Dental Sciences, Kagoshima 890-8520, Japan; arigami@m.kufm.kagoshima-u.ac.jp; 3Department of Surgery, Jiaikai Imamura General Hospital, Kagoshima 890-0064, Japan; uenosono@m3.kufm.kagoshima-u.ac.jp; 4Department of Surgery, Gyokushoukai Kajiki Onsen Hospital, Aira 899-5241, Japan; s-natsugoe@gyokushoukai.com

**Keywords:** liquid biopsy, circulating tumor cell, diagnostic value, prognostic value, esophageal cancer

## Abstract

**Simple Summary:**

The “liquid biopsy” is a novel concept for detecting circulating biomarkers in the peripheral blood of patients with various cancers, including esophageal cancer. There are two main methods to identify circulating cancer related biomarkers such as morphological techniques or molecular biological techniques. There are some differences in the sensitivity and specificity for detecting circulating tumor cells (CTCs) or circulating markers between each method. Although it is still challenging to determine strong candidates for early diagnosis and predicting prognosis in patients with esophageal cancer, our meta-analysis might be a milestone for the future development of liquid biopsies in use with esophageal cancer.

**Abstract:**

Esophageal cancer is among the most aggressive diseases, and circulating tumor cells (CTCs) have been recognized as novel biomarkers for various cancers over the past two decades, including esophageal cancer. CTCs might provide crucial clinical information for predicting cancer prognosis, monitoring therapeutic responses or recurrences, or elucidating the mechanism of metastasis. The isolation of CTCs is among the applications of a “liquid biopsy”. There are various technologies for liquid biopsies, and they are classified into two main methods: cytometric or non-cytometric techniques. Here, we review a total of 57 eligible articles to summarize various technologies for the use of a liquid biopsy in esophageal cancer and perform a meta-analysis to assess the clinical utility of liquid biopsies as a prognostic and diagnostic biomarker technique. For prognostic evaluation, the pooled hazard ratio in the cytometric assay is relatively higher than that of the non-cytometric assay. On the other hand, a combination of multiple molecules, using a non-cytometric assay, might be a favorable biomarker technique for the early diagnosis of esophageal cancer. Although determining strong evidence for a biomarker by using a liquid biopsy is still challenging, our meta-analysis might be a milestone for the future development of liquid biopsies in use with esophageal cancer.

## 1. Introduction

Esophageal cancer (EC) is the sixth leading cause of cancer-related deaths worldwide, and it can be classified into two main histological subtypes: Esophageal adenocarcinoma (EAC) and esophageal squamous cell carcinoma (ESCC). Almost all EACs develop in the lower third of the esophagus and originate from the Barrett mucosa, and ESCC occurs in the upper two-thirds of the esophagus. Approximately 80% of ESCC cases occur in central and southeastern Asia, while the incidence of EAC is high in northern and western Europe, North America, and Oceania [1]. Overall survival (OS) is similar for ESCC and EAC. Although clinical diagnostic instruments such as endoscopy, computed tomography (CT), and positron emission tomography (PET) have been developed, over sixty percent of patients are diagnosed with EC with advanced progression of the disease because of their poor symptoms in the early stage [2]. Patients with EC have poorer prognosis than those with other cancers, and even if they receive curative surgical resection, some experience early recurrence or metastasis [3]. CT and PET are still standard imaging examinations for the diagnosis and monitoring of cancers, while the limitation of low sensitivity for small lesions has been a difficult issue to detect early recurrence sites. Similarly, serum biomarkers, such as carcinoembryonic antigen (CEA) and squamous cell carcinoma antigen (SCC), have low sensitivity and specificity for early diagnosis or the detection of recurrence [4]. Therefore, a “liquid biopsy”, which is a simple and non-invasive sampling of non-solid biological tissue or DNA/RNAs from peripheral blood, is needed to provide new alternative serum biomarkers for monitoring the malignant behavior of cancer.

Ashworth et al. demonstrated the presence of circulating tumor cells (CTCs) in the peripheral blood of cancer patients in 1869 [5]. CTCs shed into the peripheral blood stream via primary tumors and extravasate into distant organs to form metastases. In the last two decades, CTCs have been identified as novel biomarkers to elucidate the mechanism of spreading metastasis and the dissemination of cancer. Allard et al. suggested that CTC measurement using the CellSearch system (Menarini-Silicon Biosystem, Bologna, Italy) might have clinical utility for all cancers of an epithelial origin [6]. To date, many kinds of liquid biopsy technologies have been reported. The most common methods of detecting CTCs are cytometric-based fluorescence immunohistochemical staining (F-IHC) and polymerase chain reaction (PCR) methods.

Quantitative reverse transcription-PCR (qRT-PCR) is a promising technique for quantifying the copy number of mRNA of interest, such as CEA, SCC, and survivin [7,8,9]. Recently, due to the rapid technological advances in molecular biology, CTC analysis has moved to the next stage, which is comprehensive analysis using microarrays or next-generation sequencing (NGS) for genetic variations. These analyses can target hundreds to thousands of microRNAs (miRNAs) or cell-free DNAs/RNAs in peripheral blood with a single procedure.

On the other hand, cytometric detection technologies have been developed that enable the capture of CTCs as a single cell and visual evaluation for phenotype characterization. CTC identification relies on positive or negative selection by leukocyte depletion. The CellSearch system, the only CTC technology cleared by the US Food and Drug Administration (FDA), is among the most representative pieces of equipment for positive selection methods. The clinical utility of CellSearch to predict tumor progression and prognosis has been reported in many cancers, including breast cancer [10], colorectal cancer [11], prostate cancer [12], and ESCC [13]. Additionally, other technologies have been reported as cytological detection methods: isolation by size of epithelial tumor cells (ISET) [14], ScreenCell [15], and MetaCell [16] as size-based separation systems; magnetic cell separation system (MACS) [17], CTC-Chip [18], and IsoFlux [19] as microfluidic-based, immune-magnetic, positive selection methods; RosetteSep as a density gradient centrifugation method [20]; and also flow cytometry. Recently, the genomic analysis of CTCs using deep sequencing was reported because of advances in single-cell analysis [21,22]. According to these studies, the detection and isolation methods of CTCs or cancer-related genes were called “liquid biopsy” methods. Many studies have attempted to demonstrate the diagnostic or prognostic value of liquid biopsies in EC, and the clinical usefulness of liquid biopsies remains controversial. Herein, we summarize articles published about liquid biopsies in EC and discuss the potential utility of liquid biopsies in clinical use.

## 2. Results

### 2.1. Literature Extraction

We identified 804 articles as potentially relevant studies, of which a total of 57 articles were finally retrieved as eligible studies according to the above criteria for this systematic review (Figure 1). A total of 11,102 cases were involved in this study, and the median number of patients was 93. In these studies, 37 articles were included in the prognostic meta-analysis and 28 articles provided diagnostic information. These studies were conducted in ten countries (Australia, China, France, Germany, Japan, Korea, Poland, Czech Republic, Taiwan, and the Netherlands), and different technologies were used to assess the utility of liquid biopsies.

### 2.2. Summary of Different Technologies for Liquid Biopsy

Currently, there are two main methods for liquid biopsies. One of them is cytological approaches to detect and isolate circulating tumor cells, and the other method involves non-cytological approaches, such as qRT-PCR, enzyme-linked immunosorbent assay (ELISA), and liquid chromatography–mass spectrometry (LC–MS) techniques to detect circulating genomic materials or metabolites from cancers. In the early 2000s, the mainstream liquid biopsy techniques were PCR for mRNAs and ELISA for proteins. A PCR assay permits the molecular detection of cancer-related mRNA expression. Moreover, qRT-PCR is a promising tool for quantifying mRNA copy numbers, and its high sensitivity is a great advantage for CTC assessment. Cytokeratin (CK), CEA, SCC, and survivin are the most common target genes for CTC in EC. Survivin is a member of the inhibitor of apoptosis genes and contributes to the inhibition of apoptosis in tumor progression [23]. However, several investigators have identified some limitations in the clinical use of PCR-based assays. False positive results are among the most important limitations, caused by cancer heterogeneity or the contamination of normal epithelial cells.

In the middle of the 2000s, cytological CTC detection technologies emerged, such as ISET and CellSearch, and not only identification but also the enumeration and characterization of CTCs became a secondary mainstream purpose in liquid biopsy studies. In the early stages, a strategy based on epithelial markers (epithelial cell adhesion molecules (EpCAMs) and CK) was used. Subsequently, the epithelial–mesenchymal transition (EMT) has been recognized as a promising phenomenon involved in the metastatic process of cancers. In the EMT, epithelial cancer cells lose their polarity and are shed into the peripheral blood circulation. To detect these EMT cells, numerous platforms with non-EpCAM-based technologies for CTC study have been developed around the world. In the 2010s, microfluidic-based platforms emerged, and combination analysis with next-generation sequencing technology developed the single-cell genomic characterization of CTCs to the next stage.

### 2.3. Summary of Platforms for the Cytometric Isolation of CTCs

#### 2.3.1. Immuno-Magnetic Technology

The CellSearch System was introduced in 1999 and is the first and only clinically validated, FDA-cleared system for the identification, isolation, and enumeration of CTCs from peripheral blood [6]. CellSearch isolates epithelial cells from peripheral blood using anti-EpCAM antigen-coated nano-ferrofluid, and CTCs are then distinguished from leukocytes by multi-fluorescent immunohistochemical staining using pan-cytokeratin (CK8, CK18, and CK19) dye 4, 6-diamidino-2-phenylidole dihydrochloride (DAPI), and CD-45 using the CellSearch Epithelial Cell Kit. These CTC separation and staining processes are fully automated by the CellTracks AutoPrep System, and CTC identification and enumeration are semi-automated by the CellTracks Analyzer II fluorescence-based microscopy system. With this system, users are able to add one more FITC (Fluorescein isothiocyanate)-labeled fluorescent reagent, such as HER2/neu antibodies. Several investigators have demonstrated that CellSearch is a promising tool for clinical management in patients with EC [13,24,25,26].

In contrast to CellSearch, CD45-labeled magnetic beads have been used for the negative selection of CTCs by the depletion of CD45-positive leukocytes. Qiao et al. reported the clinical utility of the subtraction enrichment of CTCs using CD45-coated magnetic beads (Miltenyi Biotec, Bergisch Gladbach, Germany) and triple immunohistochemical staining (CK8/18/19, CD45, and DAPI) [27]. Moreover, Li [28] and Zhang [29] demonstrated that the isolation and characterization of CTCs by CD45 bead negative selection is able to assess the chromosome multiploidy of CEP8, suggesting clear potential in improving the management of EC in clinical practice.

Another technology based on immunomagnetic CTC capture is the magnetic cell separation system (MACS) [17]. Here, cells stained sequentially with specific biotinylated antibodies, streptavidin-fluorochrome conjugates, and biotinylated superparamagnetic beads (approximately 100 nm in diameter) are separated on high-gradient magnetic columns. Unlabeled cells pass through the column, while labeled cells are retained. The retained cells can be easily eluted, and these cells are finally sorted by fluorescence microscopy or flow cytometry. MACS is also the most historical platform of CTC studies and can be applied to both positive and negative selection sequencing for downstream analysis.

Although positive enrichment depending on antibody specificity ensures high cell purity, it has several pitfalls; for example, it is difficult to detach magnetic beads coated with specific antibodies, and this antibody collection has a selection bias for CTCs. On the other hand, negative selection has a much lower purity of CTC-positive enrichment [30].

#### 2.3.2. Filtration Technology

The ISET system is among the historical CTC platforms, and the first report on it was published by Vona in 2000 [14]. Vona et al. demonstrated that ISET could isolate CTCs by size filtration using an 8 µm pore size filter from the whole blood of hepatocellular carcinoma patients. This CTC collection method supports the differences in size and rigidity between CTCs and leukocytes [31].

Furthermore, many CTC studies using ISET have been reported for many cancers, including EC [32]. Several similar devices based on filtration have been developed, such as ScreenCell [15], MetaCell [16], and more. These filtration-based CTC capture methods are simple and offer time-cost saving protocols compared to other CTC platforms. The filtration platforms are not dependent on EpCAM expression and can capture circulating epithelial–mesenchymal tumor (EMT) cells. Moreover, filtration devices can isolate CTCs, not only as single cells but also as circulating tumor microemboli (CTM). These results allow us to perform a more in-depth genomic analysis of CTCs to help shed light on the mechanisms of cancer progression.

#### 2.3.3. Microfluidic Technology

The CTC chip is the first microfluidic device to isolate CTCs, developed by Nagrath in 2007, and is based on laminar flow conditions with EpCAM-coated microposts. The CTC chip has successfully identified CTCs in the peripheral blood of patients with metastatic lung, prostate, pancreatic, breast, and colon cancer in 115 of 116 (99%) samples, with a range of 5–1281 CTCs per mL, and approximately 50% purity [18]. The CTC chip allowed us to apply subsequent analyses, such as a mutational study of genes of interest. Ohnaga reported that the CTC chip was able to isolate EC cell lines spiked in whole blood, and that its recovery rate was over 70% [33]. Following the CTC chip, many kinds of microfluidic platforms have been developed, such as ClearCell FX, based on size, deformability, and inertia separation [34]; CTC-iChip, based on positive and negative enrichment with size-based separation [35]; IsoFlux, based on automated continuous flow with EpCAM beads [19]; magnetic sifter, based on vertical flow configuration [36]; and vortex, based on inertial microfluidics and laminar microscale vortices without a red blood cells (RBC) lysis buffer [31].

In a study of esophageal cancer, two automated commercial systems have been reported. Among these is the ClearCell FX1 system, using the CTChip. This microfluidic biochip isolates CTCs based on their size, deformability, and inertia relative to other blood components using the inherent Dean vortex flows present in curvilinear channels after RBC lysis is performed. Through this process, blood cells separate and distribute themselves within the channels of the CTChip, with larger cells collecting along the inner wall and the smaller cells presenting as more separated from it. The other is the IsoFlux platform, which includes four fluidic reservoirs (sample inlet, isolation zone, waste well, and recovery tube) interconnected by microfluidic channels. Before applying samples to IsoFlux, a coupling step with immunomagnetic beads coated with anti-EpCAM antibodies is required. Samples conjugated with magnetic beads flow through the channels at a continuous flow, and each sample runs through the channel for 45 min. The isolated target cells are recovered from the isolation zone disk, which consists of a removable, low-adherence polymer disk below a magnet, and these cells are removed into a microfuge tube for further processing.

Another microfluidic platform is the fluid-assisted separation technique (FAST) disc. The FAST disc is a centrifugal and size-selective microfluidic system. This method requires no sample treatment step and the disk, passivated with bovine serum albumin, is able to isolate CTCs on the membrane by the original spin program. Finally, multi-immunofluorescent staining is performed on the FAST disc to identify CTCs [37].

### 2.4. Summary of Platforms for the Non-Cytometric Isolation of CTCs

#### 2.4.1. ELISA

ELISA is among the most common and historical protein measurement methods and has been used to measure circulating antigens and antibodies in peripheral blood [38]. Several authors have reported the clinical utility of ELISA in detect circulating cancer-related molecules. In 2005, Nozoe reported that a decreased CD4/CD8 ratio, as well as increased CD8 and decreased CD4 levels in peripheral blood, could predict worse prognosis in patients with ESCCs [39]. In 2008, Kimura demonstrated that preoperative circulating VEGF-C levels predicted recurrence in patients with EC [40].

Recently, immune checkpoint inhibitors (ICIs), such as anti-PD-1 (Programmed cell Death 1) and anti-CTLA 4 (cytotoxic T-lymphocyte-associated protein 4), have emerged as new targets for cancer treatment, and many clinical studies have been conducted. Further ELISA studies on circulating ICIs, including EC patients, are required to develop this field [41].

#### 2.4.2. RT-PCR

The PCR-based analysis is among the most common and simple techniques to identify the amplification or depletion of circulating cancer-related markers. In many cancers, including EC, the prognostic and diagnostic value of CTCs using PCR-based analysis has been reported. The greatest advantage of the PCR assay is its high sensitivity for detecting targeted circulating molecules, and we have previously investigated the clinical importance of circulating carcinoembryonic antigen (CEA) mRNA. The sensitivity, specificity, positive predictive value, and negative predictive value of CEA mRNA were higher than serum CEA or SCC, and the examination of CEA mRNA in peripheral blood during follow-up is useful for the early detection of occult recurrence [42]. The other major targets of circulating mRNA are CK19, SCC, and survivin [4]. Survivin is a member of the apoptosis inhibitor gene family and plays an important role in tumor progression. Survivin controls tumor apoptosis, promotes proliferation, and enhances angiogenesis via a vascular endothelial growth factor signaling pathway [43]. Moreover, Hoffmann demonstrated that the elevation of survivin levels after curative resection is associated with shorter OS in EC [44]. However, several investigators have noticed some limitations for PCR-based CTC analysis because the false positive results may have occurred because of the non-specific expression of targeted genes in normal and epidermal cells [45]. Therefore, further studies are needed to resolve this problem.

#### 2.4.3. Non-Coding RNAs

MicroRNAs (miRNAs) are small, non-coding single-stranded RNAs that range from 20 to 25 nucleotides in length. MicroRNAs are able to exist stably in cell-free environments such as urine and peripheral blood. The first report that investigated the potential of miRNAs as diagnostic markers was published in 2008 [46]. There are abundant studies on circulating miRNAs in many cancers, including EC, and several researchers have combined several miRNAs to improve their diagnostic value [47,48,49,50,51,52,53,54,55,56,57,58,59,60,61,62,63,64,65,66,67,68]. Currently, molecular science technology has been constantly evolving, and microarray and NGS platforms have made it possible to perform comprehensive gene analysis using liquid biopsies [53,59,60,65,66,68,69]. Many studies have investigated the clinical utility of miRNAs as diagnostic or prognostic indicators in EC patients; however, these results are still controversial due to the heterogeneity of each study caused by differences in the backgrounds and detection methods.

Long non-coding RNAs (lncRNAs) are defined as transcripts with lengths exceeding 200 nucleotides that are not translated into proteins [70]. LncRNAs play crucial roles in tumor initiation, progression, and metastasis by regulating oncogenic or tumor-suppressing pathways [71]. Similar to miRNAs, several lncRNAs have been identified as potential biomarkers in many solid cancers. There are, however, only a few studies that have been reported in EC patients. Luo et al. demonstrated that the upregulation of circulating lncRNA SNHG1 has potential as a diagnostic marker and indicates poor prognosis in ESCC [63]. Llnc-POU3F3 in plasma has also been suggested as a novel biomarker for early ESCC diagnosis [72]. Hu reported the clinical utility of three circulating lncRNAs (Linc00152, CFLAR-AS1, and POU3F3) as predictors of early ESCC progression [73].

#### 2.4.4. Circulating Tumor DNA

The presence of circulating cell-free DNA (ccfDNA) in human peripheral blood was already reported in 1948 [74], and several researchers have demonstrated that ccfDNA levels in cancer patients are elevated compared to healthy controls. In general, increased ccfDNA levels in blood correlate with unusually high cell death, linked to different pathological conditions of tumorigenesis. Such DNA fractions from tumor sites are also known as circulating tumor DNA (ctDNA). The concentration of ctDNA is extremely low, and the development of detection methods has been challenging. The simplest technique of ccfDNA quantification is concentration reading with a fluorometer. Ko demonstrated that increasing ccfDNA concentration levels during two cycles of chemotherapy can predict early disease progression and poor outcomes [75]. On the other hand, the quantification of total ccfDNA presents a risk for the contamination of non-specific ccfDNA from normal tissue, which causes poor sensitivity and specificity for the diagnosis or prediction of cancer-related outcomes. As mentioned above, new science technologies, such as NGS platforms or digital droplet PCR (ddPCR), enable ultra-sensitive analysis to detect ctDNAs and understand the characteristics of ctDNAs via somatic mutation analysis [76]. However, ccfDNA analysis has one limitation for clinical use because of the contamination from genomic DNA in the early hours after blood collection [77].

#### 2.4.5. LC–MS

Metabolomics is the systemic study of endogenous small molecule compounds (<1000 Da) in biological specimens, including blood, urine, and other tissues. Metabolomics might provide information about oncogenesis or tumorigenesis. Recently, LC–MS and nuclear magnetic resonance spectroscopy have been applied to metabolomics as novel and sensitive techniques to identify new cancer-related biomarkers. Using these platforms, several authors have demonstrated the clinical utility of metabolite combinations to detect EC in the early stages or to make differential diagnoses between high-grade dysplasia/Barrett esophagus and EC [78,79,80]. Thus, mass spectrometry-based analysis appears to be a powerful technique for biomarker detection; however, no universal instrument or method exists yet, and we are still in the process of metabolomics profiling to understand the interactions between metabolites, as well as other related molecular profiling [81].

This is why many investigators have used combinations of multiple metabolites as candidates for cancer diagnosis.

### 2.5. Prognostic Value of CTC Identification

In this section, we evaluate the prognostic values using forest plot analysis for the hazard ratio (HR) of each molecule for a “liquid biopsy”.

#### 2.5.1. OS in the Cytometric Assay

Eleven studies were enrolled for OS analysis, with a total of 854 EC patients [13,25,26,27,29,32,54,75,82,83,84,85]. In this cohort, the average age was 63.6 years (61.5–66) and the median CTC positivity rate was 46.4% (18.0–79.7). The median CTC detection rates in stages I-II and III-IV were 20.0% (0.0–33.3) and 43.5% (8.3–69.0), respectively. The median CTC detection rates in T 1–2 and 3–4 were 28.8% (7.5–55.6) and 36.2% (8.0–60.8), respectively (Table 1).

The pooled HR for OS was 2.43 (95% CI = 1.66–3.55), and poor OS was observed in CTC-positive patients compared to CTC-negative patients (Figure 2a). The negative selection with CD45 magnetic beads [27] appeared to exhibit the most powerful prognostic predictive value. There were no significant differences between each detection method because of the heterogeneity of this cohort (*I*^2^ = 59%, *p* < 0.01) and the existence of a publication bias was visualized as a funnel plot (Figure 2b).

#### 2.5.2. OS in the Non-Cytometric Assay

A total of fifteen studies were enrolled for the OS analysis of 1721 EC patients [39,44,47,48,50,52,53,54,55,73,75,88,89,90,91]. In this cohort, the average age was 61.5 years (54.3–68.7) and the median CTC positivity rate was 32.2% (22.6–77.0). The median CTC detection rates in stages I–II and III–IV were 43.2% (20.9–70.3) and 60.2% (28.4–64.6), respectively. The median CTC detection rates in T 1–2 and 3–4 were 34.5% (7.7–45.5) and 39.4% (25.4–62.0), respectively (Table 2).

The pooled HR for OS was 1.98 (95% CI = 1.56–2.51), and poorer OS was observed in CTC-positive patients compared with CTC-negative patients (Figure 2c). A combination of miRNA-3935 and miRNA-4286 [53] appeared to exhibit the most powerful prognostic predictive value. There were no significant differences between each detection method because of the heterogeneity of this cohort (*I*^2^ = 79%, *p* < 0.01) and the existence of publication bias was visualized as a funnel plot (Figure 2d).

### 2.6. Progression-Free Survival (PFS) in the Cytometric Assay

A total of ten studies were enrolled for the PFS analysis of 822 EC patients [13,25,26,27,29,54,75,84,86,87]. In this cohort, the average age was 63.3 years (61.5–66) and the median CTC positivity rate was 46.2% (18.0–79.7). The median CTC detection rates in stages I–II and III–IV were 21.1% (0.0–31.0) and 32.0% (21.3–69.0), respectively. The median CTC detection rates in T 1–2 and 3–4 were 20.0% (0.0–36.4) and 28.0% (25.5–60.8), respectively (Table 1).

The pooled HR for PFS was 2.31 (95% CI = 1.57–3.40), and poorer OS was observed in CTC-positive patients compared to CTC-negative patients (Figure 3a). The negative selection with CD45 magnetic beads [27] and positive selection with ISET [86] appeared to give the most powerful prognostic predictive value. There were no significant differences between each detection method because of the heterogeneity of this cohort (*I*^2^ = 62%, *p* < 0.01), and publication bias was visualized as a funnel plot (Figure 3b).

### 2.7. PFS in the Non-Cytometric Assay

A total of ten studies were enrolled for the PFS analysis of 1712 EC patients [8,39,40,42,47,49,51,52,54,55,75,89,90,92,93,94]. In this cohort, the average age was 62.3 years (58.9–66), and the median CTC positive rate was 35.5% (16.8–54.2). The median CTC detection rates in stages I–II and III–IV were 20.3% (15.3–38.1) and 55.9% (18.6–64.6), respectively. The median CTC detection rates in T 1–2 and 3–4 were 21.1% (7.7–40.3) and 48.1% (16.7–65.4), respectively (Table 2).

The pooled HR for PFS was 1.60 (95% CI = 1.19–2.16), and no significant difference was observed between CTC-positive and CTC-negative patients (Figure 3c). Overall, miRNA-21 [49] and SCC mRNA [8] appeared to exhibit the most powerful prognostic predictive value. There were no significant differences between each detection method because of the heterogeneity of this cohort (*I*^2^ = 85%, *p* < 0.01), and the existence of publication bias was visualized as a funnel plot (Figure 3d).

### 2.8. Early Diagnostic Value of “Liquid Biopsy”

Twenty-six molecules from twenty studies demonstrated the diagnostic value of using a liquid biopsy by using the area under the curve (AUC) for the early detection of ECs [49,50,54,56,57,58,60,61,62,63,64,65,66,67,72,75,78,79,80,95,96,97] (Table 3). It is still challenging to detect early EC or make a differential diagnosis between high-grade dysplasia and EC using a liquid biopsy strategy. In Figure 4a, the AUC values have been plotted, featuring 95% CIs. The median AUC of all the studies was 0.781 (0.550–0.991). In Figure 4b, a strong heterogeneity of publication bias is shown due to the low number of studies. Therefore, we evaluated the pooled AUC value using the random effect model, and the pooled AUC was 0.79 (0.75–0.83). These results confirmed that the copy number of ctDNA (AUC = 0.99, 95% CI = 0.98–1.00) [95], combination of metabolites (AUC = 0.96, 95% CI = 0.94–0.99) [80], and the combination of miRNAs (miR-30a-5p, 205-5p, and 574-3p) (AUC = 0.95, 95% CI = 0.90–1.00) [64] seemed to be a favorable candidate for the early detection of ECs.

Next, the derived sensitivity and 1-specificity values from each report (a total of thirty-two molecules from 22 studies [49,50,54,56,57,58,60,61,62,63,64,65,66,67,72,75,78,79,80,95,96,97]) were plotted (Figure 4c). The logarithmic regression curve, with 95% CIs, was constructed to evaluate the heterogeneity of each diagnostic value, and three molecules (lncSNHG1 [63], miR-216a [67], and a combination of miRNAs (miR16-5p, miR197-5p, and miR92a-3p) [58]) were considered as favorable candidates for the early detection of ECs.

## 3. Discussion

CTCs are unfavorable cells that are shed into the peripheral blood stream from primary tumors that can develop via metastasis. CTCs have been identified in many cancers, and their malignant behavior has been extensively demonstrated. Several investigators have reported the clinical importance of CTC in managing treatment strategies and predicting prognosis in many solid cancers, including EC. However, compared to other cancers, the clinical impact of CTCs in EC is still unclear due to the lower number of published studies. In the early 2000s, circulating mRNAs in peripheral blood were focused on as a new biomarker for the early diagnosis and prognostic prediction of various cancers. The PCR-based detection of cancer-related genes, such as CEA mRNA, CK mRNA, SCC mRNA, and survivin mRNA, presents good sensitivity in terms of predicting poor prognosis; however, some investigators have noticed that the false positive results from normal epithelial cells might constitute contamination. Since the late 2000s, cytometric methods that could morphologically identify CTCs and count the number of CTCs have been developed and have improved detection specificity. Moreover, we analyzed the genetic and mutational characteristics of each CTC. This has provided a strong contribution to clarifying the metastatic mechanisms of cancer. Although the CellSearch system is the only CTC platform that has been cleared by the FDA, several studies, including our previous study, have demonstrated the clinical utility of CellSearch as a prognostic predictor; however, the detection rate of EC, ranging from 18.0% to 50.0%, was not as high as we expected. Among the reasons for this result was that the isolation procedures of CellSearch depended on EpCAM expression. Therefore, the other non-EpCAM-dependent platforms have demonstrated a higher detection rate than CellSearch, ranging from 25.6% to 79.7%. In addition, several investigators have suggested that the measurement of the DNA methylation of cancer-related specific genes and exosomes may be used to detect early cancer or predict prognostic and therapeutic responses for several types of cancer. However, few studies have reported on DNA methylation and exosomes in esophageal cancer, and DNA methylation-based epigenetic signatures are considered to be valuable cancer biomarkers [98,99,100,101]. Nevertheless, both cytometric and non-cytometric methods could isolate and analyze cancer-related molecules or cancer cells in the peripheral blood, and these procedures have been called “liquid biopsies”, which can be performed repetitively with usual blood sampling.

In the 2010s, owing to the development of scientific technology, comprehensive gene analysis with microarrays was applied to identify circulating cancer-related non-coding RNAs, such as miRNAs and lncRNAs, and NGS was applied to sequence ctDNA. Numerous studies have been reported for many cancers, including EC, and this has become the next standard method for liquid biopsies.

In this meta-analysis, we demonstrated the pooled hazard ratio of the OS and PFS for both cytometric and non-cytometric assays. For OS, the pooled HR of the cytometric assay was relatively higher than that of the non-cytometric assay. For PFS, the pooled HR of the cytometric assay was relatively higher than that of the non-cytometric assay. It seems that the cytometric assay may be a more useful prognostic method than the non-cytometric assay; however, these results depended on the difference of each detection theory and its sensitivity. For both the cytometric and non-cytometric assay, the median CTC detection rates in stages I–II were lower than III–IV. On the other hand, the detection rate for the non-cytometric assay was relatively higher than that of the cytometric assay. These results led the prognostic value of the non-cytometric assay to be relatively lower.

Among the challenging tasks of liquid biopsies is their application to the early diagnosis of cancers. Several investigators have examined the diagnostic value of liquid biopsies for differential diagnosis between pre-cancerous diseases and early EC. Although the pooled AUC for the early diagnosis of EC was 0.79 (0.75–0.83) in this meta-analysis, it was slightly lower than other cancers; for example, the pooled AUC for hepatocellular carcinoma was 0.87 (0.83–0.89) [102], 0.89 for ovarian cancer [103], and 0.88 for colorectal cancer [104]. There were also two main reasons for this, among which was the presence of heterogeneity for the meta-analysis, owing to the small number of patients in this cohort. The other was that most studies in this review used single molecules as diagnostic predictors. Another cancer meta-analysis, as described above, applied a combination of multiple molecules and the copy-number of ctDNA as a comprehensive marker (AUC = 0.99, 95% CI = 0.98–1.00), as well as a combination of miR-30a-5p, miR-205-5p, and miR-574-3p (AUC = 0.95, 95% CI = 0.90–1.00) [64], demonstrating a favorable diagnostic value. These results suggest that it might be difficult to determine a single definitive biomarker for cancer diagnosis and that exhaustive analysis will be required.

There were some limitations to this study. First, although the meta-analysis required detailed extracted data from as many publications as possible, the number of published studies according to EC and liquid biopsies was fewer than for other cancers. Due to the small number of studies, the heterogeneity became slightly larger than expected. Second, for the non-cytometric assays, differential microarray techniques were used to determine the potential of new biomarkers, and the background of the patient in each study was different. These factors may affect the variability of the prognostic and diagnostic values, even if the same molecules came from other studies. Third, this study did not deal with the pathological differences between ESCC and EAC because of the relatively small number of EAC patients. Therefore, large-scale multicenter studies with matched-pair patients are needed to more accurately estimate the diagnostic and prognostic values of liquid biopsies.

For future perspectives on liquid biopsies, a strong and simple combination of circulating molecules will be anticipated for the clinical management of patients with EC. The early diagnosis of EC using non-invasive liquid biopsies will improve the clinical outcomes of EC, and this will provide a great contribution to reducing healthcare costs. In this meta-analysis, we did not deal with CTM because of lack of sufficient published reports for EC. Umer et al. found that CTM has higher metastatic potential and resistance to apoptosis when compared to their single cell counterparts [105]. As Umer mentioned, several investigators reported the malignant behavior of CTM in other cancers. The analysis of gene mutations, not only in CTCs but also CTM, will be a new candidate for molecular targeted therapy. Alix-Panabieres et al. advocated that CTC-derived cell lines and xenograft models are promising tools for identifying new therapeutic targets and for the development of new medicines [106].

## 4. Materials and Methods

### 4.1. Literature Search Strategy

We searched relevant articles using PubMed and Embase with the keywords “esophageal cancer”, “liquid biopsy”, and “circulating tumor cells”. An additional search with Google Scholar was performed to check for other relevant publications.

### 4.2. Inclusion and Exclusion Criteria

The following inclusion criteria were used: (1) studies were written in English; (2) studies demonstrated prognostic or diagnostic value of CTC in EC; and (3) at least 15 cases were enrolled. The exclusion criteria were the following: (1) meta-analysis, review, commentary, and laboratory articles; or (2) duplicated data reported in other studies.

### 4.3. Data Extraction

Data were retrieved from the included studies by two reviewers (M.D. and A.T.). The extracted data included the following: the first author, publication year, country, number of controls, amount of blood samples, and positive rate of CTCs in each stage. For further analysis, the hazard ratio (HR) and 95% confidence intervals (CIs) for prognosis, sensitivity, and specificity for diagnosis were retrieved. Two reviewers performed literature selection independently, and any discrepancies were resolved by discussion.

### 4.4. Statistical Methods

Prognostic meta-analysis was performed to evaluate HRs and 95% CIs by forest plot analysis using the free downloaded software EZR [107]. For diagnostic meta-analysis, the forest plot for the AUC and the summary sensitivity and specificity point with summary ROC were estimated using the JMP^®^ 11.2.0 software.

In the forest plot, the error bars indicate the 95% confidence interval (CI), the heterogeneity is indicated by *I*^2^ (intuitive statistic), and P-values less than 0.05 were considered statistically significant. The random effects model was applied to estimate the pooled HR. A funnel plot was used to evaluate publication bias.

## 5. Conclusions

Our meta-analysis confirmed the diagnostic value of liquid biopsies using a molecular combination in esophageal cancer and demonstrated that the presence of CTCs is associated with poor prognosis for both OS and PFS. We believe that this study will act as a milestone for the future development of liquid biopsies alongside esophageal cancer.

## Figures and Tables

**Figure 1 cancers-12-03070-f001:**
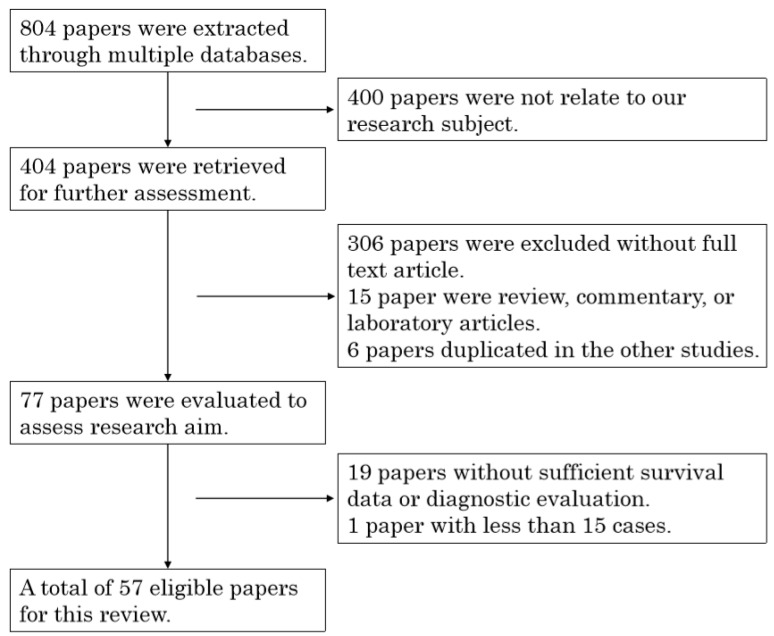
Selection process for literature in the meta-analysis.

**Figure 2 cancers-12-03070-f002:**
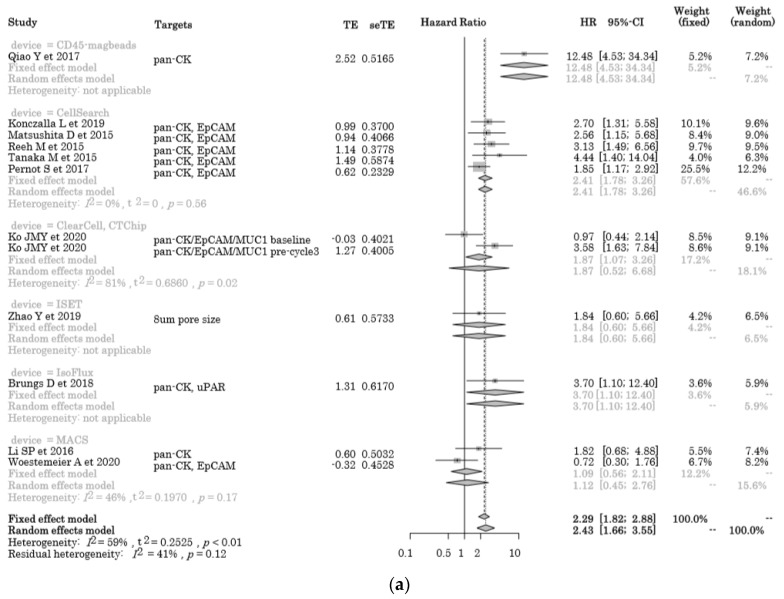
The overall survival (OS) analysis of liquid biopsies. (**a**) Forest plot of the hazard ratio for OS in the cytometric assay; (**b**) funnel plot for the publication bias of (**a**); (**c**) forest plot of the hazard ratio for OS in the non-cytometric assay; (**d**) funnel plot for the publication bias of (**c**).

**Figure 3 cancers-12-03070-f003:**
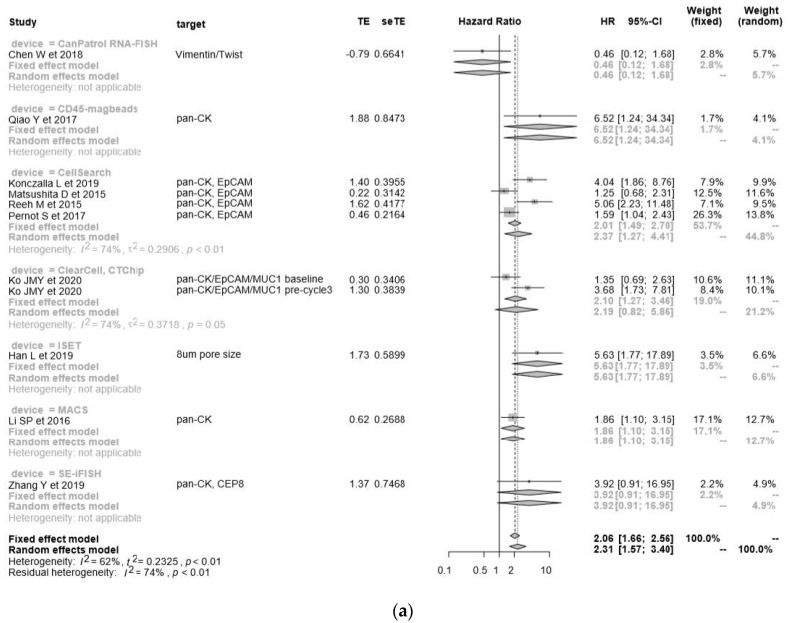
The progression free survival (FS) analysis of liquid biopsies. (**a**) Forest plot of the hazard ratio for PFS in the cytometric assay; (**b**) funnel plot for the publication bias of (**a**); (**c**) forest plot of the hazard ratio for PFS in the non-cytometric assays; (**d**) funnel plot for the publication bias of (**c**).

**Figure 4 cancers-12-03070-f004:**
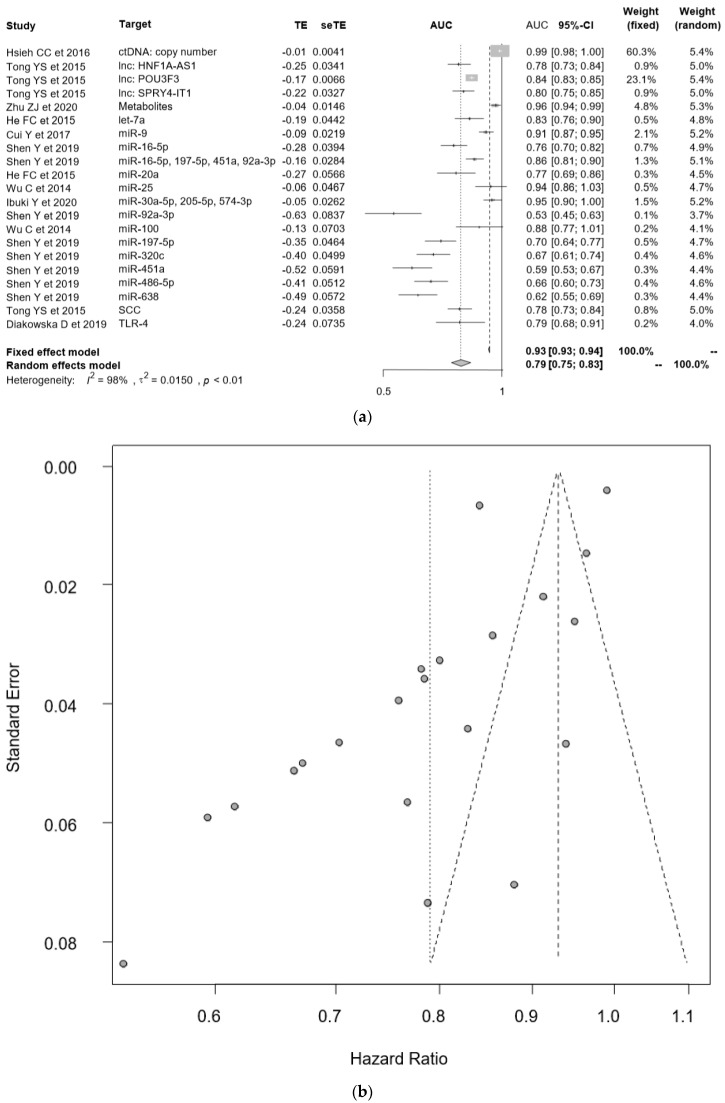
The diagnostic analysis of liquid biopsies. (**a**) Forest plot of the area under the curve; (**b**) funnel plot for the publication bias of (**a**); (**c**) the summary receiver operating characteristic (ROC) plot by sensitivity and 1-specificity for each of the molecules. The logarithmic regression curve was found to evaluate the heterogeneity of this analysis.

**Table 1 cancers-12-03070-t001:** The results of liquid biopsies for prognostic evaluation in the cytometric assays.

Author Year, Country	Technology	Molecules	SS	CS	Age (Years)	Pathology	DR	BS (mL)	DR in Stage I–II	DR in Stage III–IV	DR in T 1–2	DR in T 3–4	OS/PFS	HR	95% CIs (Hi)	95% CIs (Low)	*p*-Value
Zhao et al. [32], 2019, China	ISET	8-um pore size	55	20	-	ESCC	52.7%	5	33.3%	66.7%	55.6%	52.2%	OS	1.84	0.60	5.66	0.290
Han et al. [86], 2019, China	ISET	8-um pore size	60	0	62.2	ESCC	33.3%	5	23.5%	37.2%	0.0%	40.0%	PFS	5.63	1.77	17.89	0.003
Qiao et al. [27], 2017, China	CD45 magnetic beads	pan-CK	59	25	61.5	ESCC	79.7%	7.5	23.5%	69.0%	12.5%	60.8%	OS	12.48	8.24	34.34	0.037
													PFS	6.52	1.24	34.34	0.027
Li et al. [54], 2016, China	MACS	pan-CK	140	25	62.8	ESCC	44.3%	5	31.0%	43.5%	36.4%	51.4%	OS	1.82	0.91	4.88	0.046
													PFS	1.86	0.87	3.15	0.035
Woestemeier et al. [82], 2020, Germany	MACS	pan-CK, EpCAM	90	0	63.7	EC	25.6%	7.5	7.1%	8.3%	7.5%	8.0%	OS	0.72	0.30	1.76	0.474
Konczalla et al. [25], 2019, Germany	CellSearch	pan-CK, EpCAM	76	0	-	EC	19.7%	7.5	17.9%	21.6%	10.3%	25.5%	OS	2.70	1.31	5.58	0.007
													PFS	4.04	1.86	8.76	0.001
Matsushita et al. [13], 2015, Japan	CellSearch	pan-CK, EpCAM	90	-	65	ESCC	27.8%	7.5	0.0%	28.4%	25.0%	28.0%	OS	2.56	1.15	5.68	0.021
													PFS	1.25	0.65	2.31	0.497
Reeh et al. [26], 2015, Germany	CellSearch	pan-CK, EpCAM	100	-	66	EC	18.0%	7.5	15.1%	21.3%	32.6%	27.8%	OS	3.13	1.49	6.56	0.003
													PFS	5.06	2.23	11.48	0.001
Tanaka et al. [83], 2015, Japan	CellSearch	pan-CK, EpCAM	38	-	63	EC	50.0%	7.5	20.0%	50.0%	40.0%	44.4%	OS	4.44	1.40	14.04	0.011
Pernot et al. [84], 2017, France	CellSearch	pan-CK, EpCAM	106	0	-.	EAC	46.2%	7.5	-	-	-	-	OS	1.85	1.17	2.92	0.010
													PFS	1.59	1.04	2.43	0.030
Zhang et al. [29], 2019, China	SE-iFISH	pan-CK/CEP8	63	50	-.	ESCC	74.6%	7.5	21.1%	32.0%	20.0%	26.4%	PFS	3.92	0.91	16.95	0.047
Brungs et al. [85], 2018, Australia	IsoFlux	pan-CK/uPAR	43	0	64	EAC	46.5%	7.5	27.8%	60.0%	-	-	OS	3.70	1.20	12.4	0.030
Ko et al. [75], 2020, Korea	ClearCell, CTChip	pan-CK/EpCAM/MUC1 baseline	57	19	63	ESCC	70.9%	5	-	-	-	-.	OS	0.97	0.44	2.14	0.946
													PFS	1.35	0.69	2.63	0.380
		pan-CK/EpCAM/MUC1 pre-cycle3											OS	3.58	1.63	7.84	0.001
													PFS	3.68	1.73	7.81	0.001
Chen et al. [87], 2018, China	CanPatrol RNA-FISH	Vimentin/twist	71	40	62.7	ESCC	64.8%	5		-	-	-	PFS	0.46	0.12	1.68	0.237

SS: sample size; CS: control size; DR: detection rate; BS: blood sample; HR: hazard ratio; CI: confidence interval; ISET: isolation by size of epithelial tumor cells; MACS: magnetic cell separation system; CK: cytokeratin; EpCAM: epithelial cell adhesion molecule; OS: overall survival; PFS: progression-free survival; EC: esophageal cancer; ESCC: esophageal squamous cell carcinoma. EAC: esophageal adenocarcinoma.

**Table 2 cancers-12-03070-t002:** The results of liquid biopsies for prognostic evaluation in the non-cytometric assays.

Author, Year, Country	Technology	Molecules	SS	CS	Age (Years)	Pathology	DR	BS (mL)	DR in Stage I–II	DR in Stage III–IV	DR in T 1–2	DR in T 3–4	OS/PFS	HR	95% CIs (Hi)	95% CIs (Low)	*p*-Value
Ko et al. [75], 2020, Korea	Fluorometer	cfDNA: Baseline	57	19	63.0	ESCC	-	5	-	-	-	-	OS	8.34	2.42	28.7	0.001
													PFS	1.96	0.67	5.76	0.222
	Fluorometer	cfDNA: Pre-cycle3	57	19	63.0	ESCC	-	5	-	-	-	-	OS	5.45	1.74	17.1	0.004
													PFS	1.68	0.7	4.06	0.249
Nozoe et al. [39], 2005, Japan	ELISA	CD4/CD8 ratio	134	-	62.0	ESCC	35.80%	-	38.10%	32.00%	40.30%	31.90%	OS	1.73	1.02	2.93	0.043
													PFS	2.07	1.26	3.38	0.004
Jiao et al. [88], 2008, China	ELISA	Endothelin-1	108	82	64.5	ESCC	-	-.	-	-	-	-	OS	2.63	1.38	4.05	0.003
Blanchard et al. [89], 2012, France	ELISA	Kras	84	-	60.0	EC	22.60%	-	-	-	7.70%	25.40%	OS	0.8	0.5	1.5	0.5
													PFS	0.8	0.5	1.4	0.5
	ELISA	p53	84	-	60.0	EC	28.60%	-	-	-	7.70%	32.40%	OS	2	1.05	2.8	0.04
													PFS	2	1	3.9	0.04
Kimura et al. [40], 2008, Japan	ELISA	VEGF-C	80	20	62.8	EC	-	-	-	-	-	-	PFS	5.6	1.6	19.6	0.007
Hu et al. [73], 2016, China	qRT-PCR	lnc CFLAR-AS1	205	210	54.3	ESCC	-	-	-	-	-	-	OS	1.68	1.08	2.32	N.D.
		Linc00152	205	210	54.3	ESCC	-	-	-	-	-	-	OS	1.89	1.22	2.58	N.D.
		lnc POU3F3	205	210	54.3	ESCC	-	-	-	-	-	-	OS	1.82	1.17	2.51	N.D.
Li et al. [47], 2017, China	qRT-PCR	miR-15a	106	106	62.3	EC	-	-	-	-	-	-	OS	4.17	1.97	10.63	0.01
													PFS	4.01	1.62	9.82	0.01
Lv et al. [48], 2016, China	qRT-PCR	miR-21	126	80	59.1	ESCC	-	-	48.40%	60.90%	34.50%	62.00%	OS	1.85	1.48	6.24	0.012
		miR-375	126	80	59.1	ESCC	-	-	66.10%	34.40%	45.50%	39.40%	OS	0.65	0.27	0.86	0.041
Komatsu et al. [49], 2016, Japan	qRT-PCR	miR-21	37	20	-	ESCC	43.20%	7	-	-	-	-	PFS	9.95	1.56	63.42	0.015
Wu et al. [50], 2014, China	qRT-PCR	miR-25	194	94	-	ESCC	-	2	61.90%	62.50%	-	-	OS	1.13	0.78	1.64	0.526
		miR-25	63	63	-	ESCC	-	3	-	-	-	-	OS	3.84	1.02	14.41	0.046
		miR-100	63	63	-	ESCC	-	3	-	-	-	-	OS	4.18	1.21	14.5	0.024
		miR-223	194	94	-	ESCC	-	2	56.80%	59.40%	-	-	OS	1.72	1.14	2.59	0.01
		miR-375	194	94	-	ESCC	-	2	70.30%	64.10%	-	-	OS	1.75	1.11	2.76	0.016
Gu et al. [51], 2018, China	qRT-PCR	miR-25-3p	329	-	61.7	EAC	-	-	-	-	-	-	PFS	1.04	0.73	1.25	0.817
		miR-30c-5p	329	-	61.7	EAC	-.	-	-	-	-	-	PFS	0.86	0.61	1.22	0.397
		miR-152-3p	329	-	61.7	EAC	-	-	-	-	-	-	PFS	0.78	0.54	1.11	0.161
		miR-331-3p	329	-	61.7	EAC	-	-	-	-	-	-	PFS	0.55	0.38	0.78	0.001
Tanaka et al. [52], 2013, Japan	qRT-PCR	miR-200c	64	27	-	ESCC	-	8	28.00%	64.10%	30.00%	59.10%	OS	2.79	1.11	7.96	N.D.
													PFS	2.79	1.11	7.96	0.029
Zhai et al. [53], 2015, China	qRT-PCR	miR-3935, 4286	30	30	68.7	EC	-	-	-	-	-	-	OS	10.91	1.8	66.12	0.009
Li et al. [54], 2016, China	qRT-PCR	miR-506	100	40	59.2	ESCC	-	5	30.40%	51.90%	35.40%	48.10%	OS	2.35	1.32	4.2	0.004
													PFS	2.65	1.53	4.58	1
Guan et al. [55], 2016, China	qRT-PCR	miR-613	75	75	65.0	ESCC	-	-	-	-	-	-	OS	0.59	0.34	0.95	0.031
													PFS	0.66	0.48	0.89	0.006
Setoyama et al. [42], 2006, Japan	qRT-PCR	CEA	106	28	63.3	EC	36.80%	5	19.70%	60.00%	18.30%	56.50%	PFS	0.53	0.32	0.8	0.002
Tanaka et al. [92], 2010, Japan	qRT-PCR	CEA, SCC	244	-	-	ESCC	16.80%	8	15.30%	18.60%	17.00%	16.70%	PFS	1.65	1.03	2.63	0.037
Yin et al. [93], 2012, China	qRT-PCR	CEA, survivin, CK19	72	-	63.0	EC	54.20%	-	-	-	-.	-	PFS	3.68	1.38	9.84	0.008
Honma et al. [94], 2006, Japan	qRT-PCR	SCC	46	42	66.0	ESCC	30.40%	3	16.70%	39.30%	21.10%	37.00%	PFS	3	1.05	8.54	0.04
Kaganoi et al. [8], 2004, Japan	qRT-PCR	SCC	70	19	-	ESCC	32.80%	10	18.80%	63.60%	13.60%	65.40%	PFS	7.15	1.25	61.1	0.038
Hoffmann et al. [44], 2010, Germany	qRT-PCR	Survivin	62	-	-	EC	77.00%	-	-	-	-	-	OS	6.6	1.97	22.12	0.002
Cao et al. [90], 2009, China	qRT-PCR	Survivin	108	75	58.9	ESCC	47.20%	2	20.90%	64.60%	36.50%	57.10%	OS	5.17	2.3	11.65	0.001
													PFS	5.18	2.42	8.93	0.005
He et al. [91], 2019, China	qRT-PCR	uPA	205	-	-	ESCC	25.90%	-	22.90%	28.40%	16.70%	29.70%	OS	1.82	1.16	2.85	0.009
													PFS	1.97	1.11	3.49	0.02

SS: sample size; CS: control size; DR: detection rate; BS: blood sample; HR: hazard ratio; CI: confidence interval; CEA: carcinoembryonic antigen; SCC: squamous cell carcinoma antigen.

**Table 3 cancers-12-03070-t003:** The results of the liquid biopsies for diagnostic evaluation.

Author, Year, Country	Technology	Molecules	SS	CS	Age (Years)	Pathology	AUC	Sensitivity	Specificity	95% CIs (Hi)	95% CIs (Low)
Ko et al. [75], 2020, Korea	ClearCell CTChip	pan-CK/EpCAM/MUC1 baseline	57	19	63	ESCC	0.681	0.455	0.895		
Hsieh et al. [95], 2016, Taiwan	qRT-PCR	ctDNA: Copy number	81	95	60.4	ESCC	0.991	0.963	0.941	0.982	0.999
Liao et al. [97], 2017, China	ELISA	FAPα	151	194	62	ESCC	0.714	0.561	0.856		
Tong et al. [72], 2015, China	qRT-PCR	lnc POU3F3	147	23	-	ESCC	0.842	0.728	0.894	0.748	0.853
Luoet al. [63], 2020, China	qRT-PCR	lnc SNHG1	60	60	-	ESCC	0.85	0.774	0.925		
Zhang et al. [78], 2012, China	LC–MS	Metabolites	67	34	-	EAC	0.92	0.89	0.9		
Xuet al. [79], 2106, China	LC–MS	Metabolites	62	62	62	EC	0.981	0.913	0.984		
Zhu et al. [80], 2020, China	LC–MS	Metabolites	140	170	60	ESCC	0.965	0.883	0.889	0.936	0.993
He et al. [56], 2015, China	qRT-PCR	let-7a	70	40	60.5	ESCC	0.829	0.743	0.85	0.754	0.904
Cui et al. [57], 2017, China	qRT-PCR	miR-9	131	131	-	ESCC	0.913	0.855	0.985	0.873	0.953
Shen et al. [58], 2019, China	qRT-PCR	miR-16-5p, 197-5p, 451a, 92a-3p	96	78	60.1	ESCC	0.856	0.896	0.763	0.794	0.905
Zheng et al. [60], 2019, China	qRT-PCR	miR-16-5p, 451a, 574-5p	23	23	-	ESCC	0.76	0.73	0.82		
Hirajima et al. [61], 2013, Japan	qRT-PCR	miR-18a	106	54	-	ESCC	0.9449	0.868	1		
He et al. [56], 2015, China	qRT-PCR	miR-20a	70	40	60.5	ESCC	0.767	0.643	0.75	0.677	0.857
Zhang et al. [62], 2018, China	qRT-PCR	miR-21	125	125	63	ESCC	0.8	0.74	0.78		
Luo et al. [63], 2020, China	qRT-PCR	miR-21	60	60	-	ESCC	0.928	0.883	0.973		
Komatsu et al. [49], 2016, Japan	qRT-PCR	miR-21	37	20	-	ESCC	0.8154	0.542	0.923		
Zhang et al. [62], 2018, China	qRT-PCR	miR-25	125	125	63	ESCC	0.55	0.54	0.57		
Wu et al. [50], 2014, China	qRT-PCR	miR-25	194	94	-	ESCC	0.593	0.471	0.716		
Ibuki et al. [64], 2020, Japan	qRT-PCR	miR-30a-5p, 205-5p, 574-3p	66	42	-	ESCC	0.95	0.938	0.81	0.91	1
Zhang et al. [62], 2018, China	qRT-PCR	miR-100	125	125	63	ESCC	0.58	0.58	0.58		
Bus et al. [65] 2016, Netherlands	qRT-PCR	miR-133a-3p, 136-5p, 382-5p	59	15	65.8	EAC	0.797	0.8095	0.7838		
Pavlov et al. [66], 2018, Netherlands	qRT-PCR	miR-199a-3p, 320e	17	19	65.1	EAC	0.786	0.823	0.622		
Dong et al. [67], 2016, China	qRT-PCR	miR-216a	120	51	-	ESCC	0.877	0.8	0.902		
Dong et al. [67], 2016, China	qRT-PCR	miR-216b	120	51	-	ESCC	0.756	0.558	0.902		
Zhang et al. [62], 2018, China	qRT-PCR	miR-223	125	125	63	ESCC	0.73	0.68	0.68		
Zhang et al. [62], 2018, China	qRT-PCR	miR-375	125	125	63	ESCC	0.69	0.78	0.59		
Li et al. [54], 2016, China	qRT-PCR	miR-506	100	40	59.2	ESCC	0.835	0.8636	1		
Tong et al. [72], 2015, China	qRT-PCR	SCC	147	23	-	ESCC	0.784	0.592	0.935	0.727	0.841
Diakowska et al. [96], 2019, Poland	qRT-PCR	TLR-4	27	38	-	EAC	0.787	0.7	0.78	0.661	0.909

SS: sample size; CS: control size; AUC: area under curve; CI: confidence interval.

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
