# Peer review of "The Diagnostic and Prognostic Value of a Liquid Biopsy for Esophageal Cancer: A Systematic Review and Meta-Analysis"

_cancers, 2020, doi:10.3390/cancers12103070_

Round 1
Reviewer 1 Report
Authors conducted a systematic review and meta-analysis of diagnostic and prognostic value of liquid biopsy for esophageal cancer. Overall, this manuscript is not well written as it does not follow the correct format of performing a systematic review or meta-analysis. In particular, the the first half of the manuscript was not written well, especially from section 2.1 to 2.4 as there were not results presented in these section, more like discussion. The second half of the manuscript from section 2.5 is well presented showing the meta-analysis of these eligible studies. Thus, authors need to revise it and make sure that only results from your review and meta-analysis will be presented not focusing on discussion. Specific comments are provided below:
- Abstract: not well presented as there are no findings presented at all. Contents are more like a general literature review. Authors need to provide the number of eligible studies included in the review and analysis and key findings from your meta-analysis. Please revise it accordingly.
- Introduction: should avoid having only one sentence in a paragraph.
- Results, page 2, section 2.1, it is confusing with regard to the number of eligible studies and number of studies included in the analysis. The main text shows 69 article met the selection criteria, while figure 1 shows 57 are eligble. So what is the exact number of studies that were eligible? Also, for searching strategy, did authors follow some guidlines such as PRISMA guidelines for conducting systematic review and meta-analysis? Also, please cite the eligible studies in the main text so that readers are aware of these eligible studies.
- Results: section 2.2-2.4, poorly presented as these sections did not present any findings related to your review or analysis. it is more like a discussion. Please revise these sections. Section 2.5 onwards is well presented showing the meta-analysis.
- Methods: data were retrived by two reviewers. Did they extract the separately? how to resolve discrepany if there was any? Was the third reviewer involved to solve it? Please clarify it.
- Figures and tables are nicely presented.
Author Response
1. Abstract: not well presented as there are no findings presented at all. Contents are more like a general literature review. Authors need to provide the number of eligible studies included in the review and analysis and key findings from your meta-analysis. Please revise it accordingly.
⇒Thanks for your precise identification. We revised ABSTRACT followed your advice.
2. Introduction: should avoid having only one sentence in a paragraph.
⇒We corrected this point (line 40-41).
3. Results, page 2, section 2.1, it is confusing with regard to the number of eligible studies and number of studies included in the analysis. The main text shows 69 article met the selection criteria, while figure 1 shows 57 are eligble. So what is the exact number of studies that were eligible? Also, for searching strategy, did authors follow some guidlines such as PRISMA guidelines for conducting systematic review and meta-analysis? Also, please cite the eligible studies in the main text so that readers are aware of these eligible studies.
⇒We are sorry for the confusion, “57” is a correct number of eligible articles for review. We confirmed that there were any mistakes in the number of total cases and number of studies for meta-analysis.
4.Results: section 2.2-2.4, poorly presented as these sections did not present any findings related to your review or analysis. it is more like a discussion. Please revise these sections. Section 2.5 onwards is well presented showing the meta-analysis.
⇒Thank you for your comment. We considered that section 2.2-2.4 were summaries of this review assessing the differentiation of CTC detection methods, and we changed titles of each section (line 99, 123 and 199).
5. Methods: data were retrived by two reviewers. Did they extract the separately? how to resolve discrepany if there was any? Was the third reviewer involved to solve it? Please clarify it.
⇒Two reviewers performed literature selection independently, and any discrepancy was resolved by discussion. (We add this sentence in line 478-479.)
Reviewer 2 Report
Recommendation: Major revisions
Comments:
This manuscript describes the diagnostic and prognostic value of liquid biopsy for esophageal cancer. The authors need to address the following comments and revise the manuscript accordingly.
- This manuscript is in need of substantial editing, English language and style improvement.
- Page 1, line 22: Please consider to re-write the abstract. The manuscript describes the diagnostic and prognostic value of liquid biopsy not CTC only. Cover entire liquid biopsy.
- Page 2, line 53: Describe “liquid biopsy” briefly and the potential of liquid biopsy for early detection of cancer.
- CfDNA is one of the key components in liquid biopsy based detection. Please consider to include DNA methylation based assays for early detection of cancer. Consider to add the following references. a) Roy, D. and Tiirikainen, M. Diagnostic Power of DNA Methylation Classifiers for Early Detection of Cancer, Trends in Cancer, Volume 6, Issue 2, February 2020, Pages 78-81. b) Laird PW. The power and the promise of DNA methylation markers. Nat Rev Cancer 2003; 3(4): 253–266. c) D. Roy, D.Taggart, L. Zheng, D. Liu, G. Li, M. Li, K. Zhang, R. A. Van Etten. Circulating cell-free DNA methylation assay: Towards early detection of multiple cancer types [abstract]. In: Proceedings of the American Association for Cancer Research Annual Meeting 2019; 2019 Mar 29-Apr 3; Atlanta, GA. Philadelphia (PA): AACR; Cancer Res 2019;79(13 Suppl):Abstract nr 837. e) Duffy MJ, Napieralski R, Martens JWM et al. Methylated genes as new cancer biomarkers. Eur J Cancer 2009; 45(3): 335–346.
- Please consider to cover exosomes
Author Response
- Page 1, line 22: Please consider to re-write the abstract. The manuscript describes the diagnostic and prognostic value of liquid biopsy not CTC only. Cover entire liquid biopsy. ⇒Thanks for your precise identification. We revised ABSTRACT followed your advice
- Page 2, line 53: Describe “liquid biopsy” briefly and the potential of liquid biopsy for early detection of cancer. ⇒“Liquid biopsy” which is a simple and non-invasive sampling of non-solid biological tissue or DNA/RNAs from peripheral blood is needed to provide new alternative serum biomarkers for monitoring the malignant behavior of cancer. (added in line 57-60)
- CfDNA is one of the key components in liquid biopsy based detection. Please consider to include DNA methylation based assays for early detection of cancer. Consider to add the following references. a) Roy, D. and Tiirikainen, M. Diagnostic Power of DNA Methylation Classifiers for Early Detection of Cancer, Trends in Cancer, Volume 6, Issue 2, February 2020, Pages 78-81. b) Laird PW. The power and the promise of DNA methylation markers. Nat Rev Cancer 2003; 3(4): 253–266. c) D. Roy, D.Taggart, L. Zheng, D. Liu, G. Li, M. Li, K. Zhang, R. A. Van Etten. Circulating cell-free DNA methylation assay: Towards early detection of multiple cancer types [abstract]. In: Proceedings of the American Association for Cancer Research Annual Meeting 2019; 2019 Mar 29-Apr 3; Atlanta, GA. Philadelphia (PA): AACR; Cancer Res 2019;79(13 Suppl):Abstract nr 837. e) Duffy MJ, Napieralski R, Martens JWM et al. Methylated genes as new cancer biomarkers. Eur J Cancer 2009; 45(3): 335–346.
- Please consider to cover exosomes
⇒Thank you for your advice, we added a summary of these studies (line 417-421).
[102] Zhu Y, Zhang H, Chen N, et al. Diagnostic value of various liquid biopsy methods for pancreatic cancer: A systematic review and meta-analysis. Medicine (Baltimore). 2020 Jan;99(3):e18581.
Reviewer 3 Report
Congratulations to the authors for this very well written, interesting systematic review and meta-analysis about the value of liquid biopsy in oesophageal cancer.
There is just one minor comment that may be addressed by the authors during potential re-submission:
- A graph or table listing the different types of liquid biopsies (as described in chapters 2.2 to 2.4) would further aid readers to understand the diversity of methods available and their implication in clinical practice.
Author Response
A graph or table listing the different types of liquid biopsies (as described in chapters 2.2 to 2.4) would further aid readers to understand the diversity of methods available and their implication in clinical practice.
⇒Thank you for your advice. We had just considered to add a list as you mentioned, we think that table 1 and 2 might already summarize this point..
Round 2
Reviewer 1 Report
Authors revised the manuscript and addressed most of the comments. However, I noticed some errors in the revision, for example:
- it seems they used track changes in making the revision, but I do not understand why the original words/letters are still kept. Such as the title change: The Didiagnostic and Pprognostic... why d and p were not removed? This is observed in many other areas throughout the manuscript. Please check it again and fix these errors.
- Page 2, Introduction, first sentence, Esophageal cancer (EC) is the sixth leading cause of cancer-related deaths worldwide, and it . it? the sentence is incomplete.
Author Response
Thank you for your suggestions, and I apologize to you for my complicated revision.
Here, I corrected my revised article to be easy to read.
Reviewer 2 Report
Comments:
The authors have addressed the comments quite thoroughly and this version of the manuscript is improved. Please publish.
Author Response
I also appreciate the time and effort you have dedicated to providing insightful feedback on ways to strengthen our paper.